# Probing plexciton emission from 2D materials on gold nanotrenches

Junze Zhou [1,4] ✉, P. A. D. Gonçalves [2,4], Fabrizio Riminucci [1], Scott Dhuey[1], Edward S. Barnard [1], Adam Schwartzberg [1], F. Javier García de Abajo [2,3] ✉ & Alexander Weber-Bargioni[1] ✉

Probing strongly coupled quasiparticle excitations at their intrinsic length scales offers unique insights into their properties and facilitates the design of devices with novel functionalities. In this work, we investigate the formation and emission characteristics of plexcitons, arising from the interaction between surface plasmons in narrow gold nanotrenches and excitons in monolayer $WSe_2$. We study this strong plasmon–exciton coupling in both the far-field and the near-field. Specifically, we observe a Rabi splitting in the far-field reflection spectra of about 80 meV under ambient conditions, consistent with our theoretical modeling. Using a custom-designed near-field probe, we find that plexciton emission originates predominantly from the lower-frequency branch, which we can directly probe and map its local field distribution. We precisely determine the plexciton's spatial extension, similar to the trench width, with nanometric precision by collecting spectra at controlled probe locations. Our work opens exciting prospects for nanoscale mapping and engineering of plexcitons in complex nanostructures with potential applications in nanophotonic devices, optoelectronics, and quantum electrodynamics in nanoscale cavities.

The exploration of strongly coupled quasiparticle excitations in quantum materials has opened a new frontier in nanophotonics[1–4]. A prominent example of such excitations are plexcitons[5,6], resulting from the interaction between plasmon resonances and excitons[3,7,8]. Plexcitons have been shown to produce large optical nonlinearities[9,10], lead to ultrafast energy exchange[11], and enable investigations of quantum phenomena at room temperature[1,2,12,13], thus offering an enticing route to engineer light–matter interactions at the nanoscale. In recent years, the field has been fueled by the emergence of two-dimensional (2D) transition-metal dichalcogenides[8,14] (TMDs) as robust materials that host excitons with large binding energies and oscillator strengths[7,15,16], making them exceptionally attractive for achieving strong coupling at room temperature and, thus, facilitating the exploration of plexcitonic phenomena.

Despite intense efforts to explore plasmon–exciton coupling in monolayer TMD/plasmonic hybrid systems, the nanoscale properties of plexcitons have so far been largely unexplored. The difficulty of accessing small plasmonic cavities and the limitations of conventional, diffraction-limited photoluminescence (PL) setups have made probing these interactions extremely challenging[17,18]. While electron energy-loss spectroscopy (EELS) has been employed to visualize strong coupling at the nanoscale[19], it was restricted to multilayer TMDs, where excitonic properties are less prominent compared to monolayers, and it cannot access the emission properties effectively. Moreover, strong plasmon–exciton coupling has been primarily realized by placing plasmonic nanoparticles onto TMD-covered substrates[20–23] (including metallic ones, in the so-called nanoparticle-on-a-mirror setup[24]), making nanoscale characterization techniques based on scanning probes,

[1]The Molecular Foundry, Lawrence Berkeley National Laboratory, 1 Cyclotron Road, Berkeley, CA, USA. [2]ICFO – Institut de Ciencies Fotoniques, The Barcelona Institute of Science and Technology, Castelldefels (Barcelona), Spain. [3]ICREA – Institució Catalana de Recerca i Estudis Avançats, Passeig Lluís Companys 23, Barcelona, Spain. [4]These authors contributed equally: Junze Zhou, P. A. D. Gonçalves. ✉e-mail: junzezhou@lbl.gov; javier.garciadeabajo@nanophotonics.es; afweber-bargioni@lbl.gov

such as scanning near-field optical microscopy or scanning tunneling microscopy, nearly impossible. Thus, directly accessing and probing a plexciton in the near-field has remained a pending challenge, yet is key to advancing our fundamental understanding of plexciton physics and to the purposeful design of plexcitonic devices.

In this work, we present a combined far- and near-field investigation of plasmon–exciton coupling in WSe₂-covered gold nanotrenches. This platform enables direct access to the nanoscale properties of plexcitons and creates highly localized and easily tunable plasmon resonances that can be tailored to match the TMD exciton energy. Using near-field hyperspectral PL imaging, we unambiguously demonstrate the formation of plexcitons, their strong localization over individual nanotrenches, as well as their emission characteristics. Our findings are further supported by a comprehensive theoretical modeling of the nanoscale optical response of the plasmonic structure and of the coupled plasmon–exciton complex. Our work not only overcomes previous limitations for directly mapping plexcitonic excitations at the nanoscale, but it also provides a unique platform for in-depth explorations of the properties and emission dynamics of plexcitons that could find applications in nanophotonic devices[25], optoelectronics[26], and quantum electrodynamics in nanoscale cavities[27].

## Results

### Sample design and plexciton signature

The light–matter interaction is quantified through the coupling strength, which for a plexciton is $g = \mathbf{\mu}_e \cdot \mathbf{E}_p(\mathbf{r}_e)$, where $\mathbf{\mu}_e$ is the transition-dipole moment of the exciton and $\mathbf{E}_p(\mathbf{r}_e)$ the electric field of the plasmon at the exciton's position. Therefore, plexciton formation requires a substantial spectral and spatial overlap between the plasmon and the exciton. The coupling is facilitated when the plasmon electric field aligns with the exciton dipole moment [i.e., parallel to the TMD plane for optically active (bright) excitons[28,29]].

To investigate plexciton formation while enabling near-field probe access, we have developed plasmonic cavities consisting of one-dimensional (1D) gold nanotrenches conformally coated with 2 nm of Al₂O₃ (to prevent quenching) together with a monolayer of WSe₂ placed on top (Fig. 1a). Such plasmonic cavities support resonances that provide large field enhancements at the opening of the nanotrench, and thus, near the 2D material (Fig. 1b). Moreover, the plasmon resonance features a strong in-plane dipole oriented across the trench ($x$-axis)[30–32], which appropriately aligns with the exciton's in-plane dipole moment. We fabricate our devices using well-established techniques (see Supp. Sec. S1)[30,33], guided by simulations and reflection measurements to precisely tune the plasmon resonance to the exciton

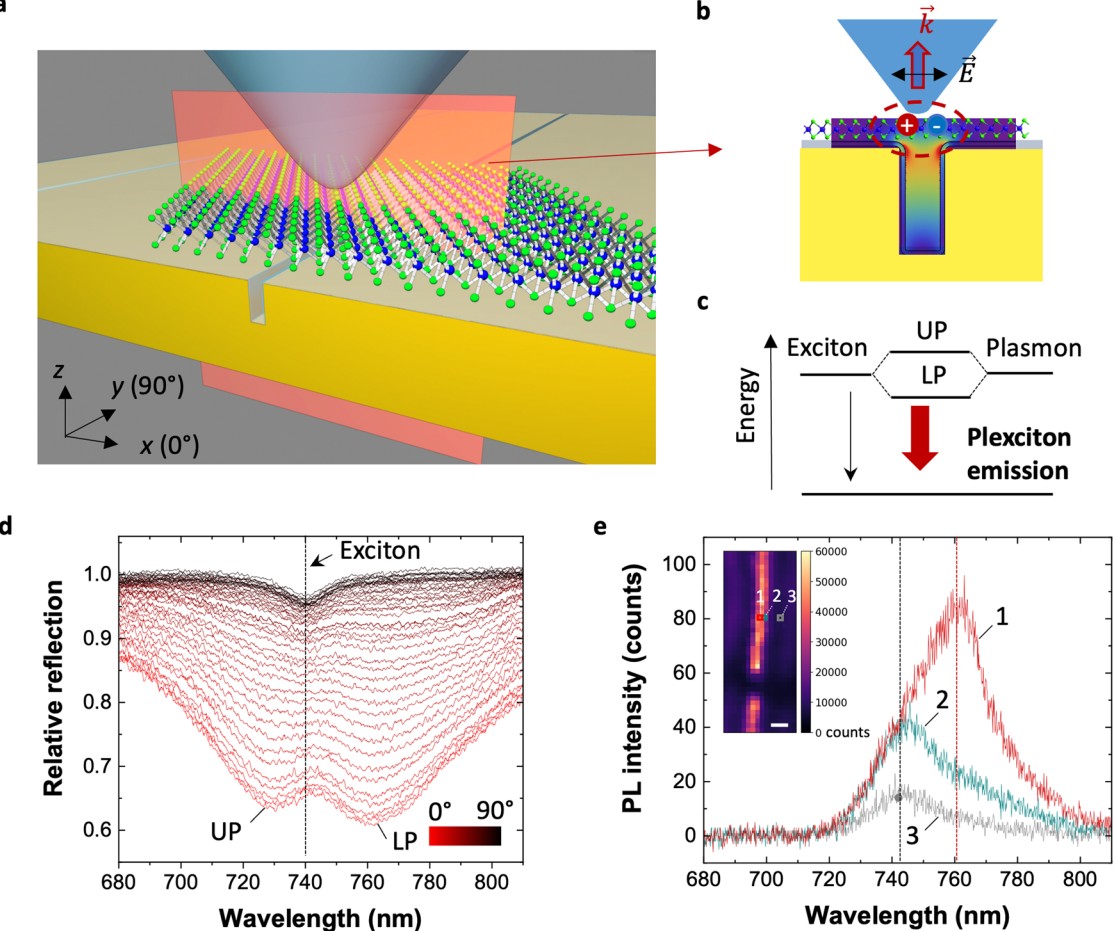

**Fig. 1 | Near-field probing and optical characterization of plexciton emission in a coupled WSe₂–Au-nanotrench system. a** Schematic representation of the device, featuring a WSe₂ monolayer (not to scale) deposited on a gold nanotrench coated with an ≈ 2-nm-thick Al₂O₃ layer. The emitted light is collected using a sharp pyramidal probe (tip curvature radius of 20 nm) integrated into fiber facets (see Supp. Sec. S2 for additional details). **b** Cross-sectional view (red plane in **a**) of the plasmon field intensity enhancement in the bare trench. **c** Energy-level diagram illustrating the plexciton system, with the thick-red arrow denoting plexciton

emission primarily originating from the lower polariton (LP), and with the thin-dark arrow representing the emission of the bare (uncoupled) TMD exciton. **d** Reflection spectrum as a function of incident-light polarization angle (0° to 90° for perpendicular-to-parallel to the trench). Each curve is normalized relative to measurements on the unpatterned substrate. **e** Photoluminescence spectra acquired with our fiber-based pyramidal probe at positions 1–3 indicated in inset (scale bar is 200 nm).

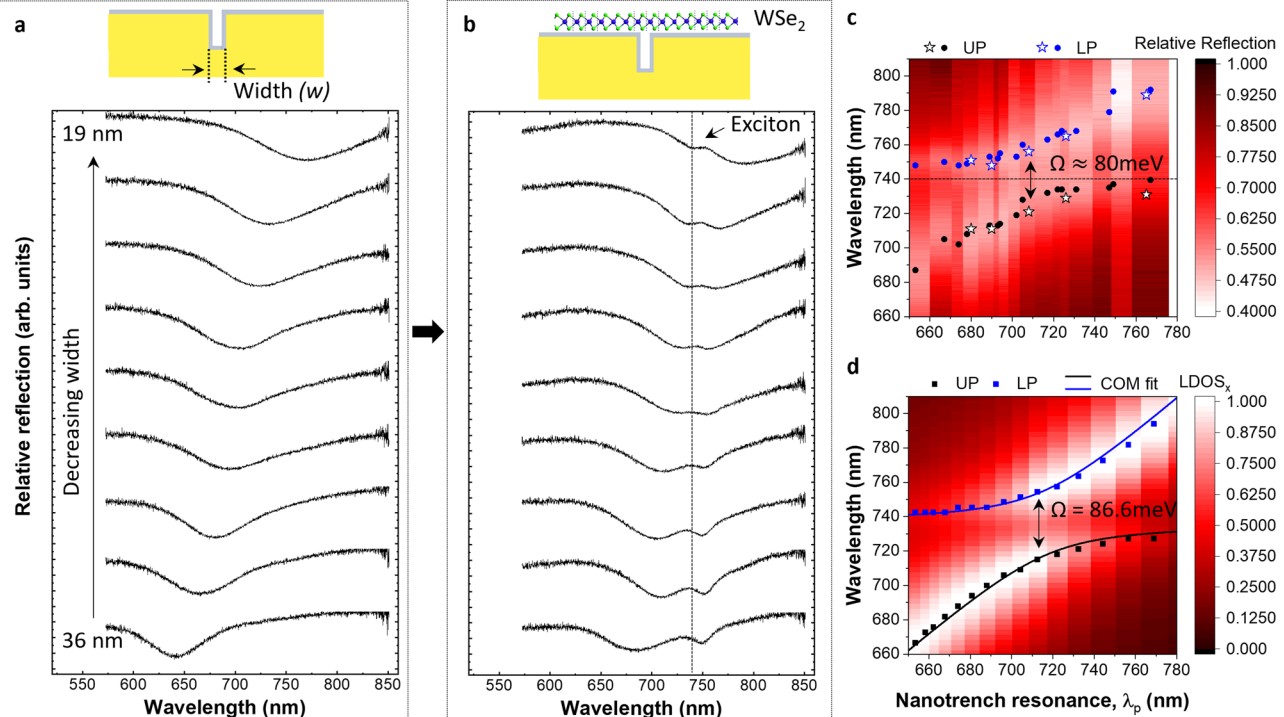

**Fig. 2 | Strong plasmon–exciton coupling. a** Reflection spectra from bare gold nanotrenches (see top schematic) of varying width, normalized to the ones from flat susbtrates. The curves are vertically offset for clarity. **b** Reflection spectra of the WSe$_2$–Au-nanotrench coupled system, following the same width order as in (**a**). **c** Wavelength dependence (vertical axis) of the UP (black dots/stars) and LP (red dots/stars) as a function of plasmon resonance wavelength (horizontal axis) identified in the uncovered nanotrench areas. The energy splitting at zero detuning is ≈ 80 meV. **d** Calculated dispersion diagram based on the *x*-resolved LDOS using the BEM (see SI for details), where the (normalized) maxima corresponding to the LP and UP branches are indicated by the blue and black squares. Fitting to a coupled-oscillator model (COM), renders a Rabi-like splitting of Ω ≈ 80 meV in experiment (**c**) and Ω ≈ 86.6 meV in the BEM calculations (see Supp. Sec. 5A).

wavelength (see Supp. Sec. S4). Under strong-coupling conditions, the plasmon and exciton hybridize forming two new plexciton states. We detect the signature of the plexciton through reflection measurements (Fig. 1d) in the far-field and via near-field photoluminescence (Fig. 1a–c, e).

In order to demonstrate that our platform provides enough strong coupling to create a plexciton, we start by performing polarization-dependent reflection measurements, illuminating the sample with linearly polarized white light, with the polarization angle varying from 0° (along the *x*-axis, perpendicular to the trench) to 90° (*y*-axis, parallel to the trench). The dips identified in the reflection spectra correspond to peaks in absorption spectra[34], which are typically observed in the far-field for strongly coupled systems. The results clearly show the presence of two well-resolved dips in the reflection data for 0° polarization, which progressively merge into a single dip at the bare exciton wavelength as the polarization is rotated to 90° (Fig. 1d). Such a behavior is a hallmark signature of strong plasmon–exciton coupling since the level splitting is only observed when the nanotrench plasmon is efficiently excited with near 0° polarization (see Supp. Fig. S19).

We then investigate the nanoscale optical response of plexcitons via scanning near-field optical microscopy (SNOM) using a fiber-based pyramidal probe fabricated with a nanoimprinting procedure[35] (see Methods). We start by collecting the near-field signal using a bare dielectric probe to (i) minimize the tip-induced perturbation of the plasmonic resonance and (ii) to prevent PL quenching, which are known issues when using metallic tips[36–39]. The probe allows us to characterize the plexciton emission via hyperspectral PL imaging by exciting the sample with a continuous wave (CW) laser at 633 nm wavelength coupled through a glass fiber (see Methods), where the light is tightly focused by the pyramidal probe at the end of the fiber into a nearly

diffraction-limited spot[35]. The plexcitonic PL emission is recorded by raster-scanning the probe in the *xy*-plane, exhibiting enhanced emission at the nanotrench location indicated by the bright emission stripe in the inset in Fig. 1e. The inset shows the integrated PL intensity at each pixel (for the correlated height map, see Supp. Fig. S5c). The spectrum captured at the nanotrench (position 1 in the inset) displays a prominent peak at 763 nm (indicated by the vertical red-dotted line in Fig. 1e), which is consistent with the lower polariton (LP) absorption identified in Fig. 1d. We did not detect any observable PL emission from the upper polariton (UP) branch, likely because it is quenched by fast nonradiative relaxation, in agreement with previous strong-coupling studies[14,22]. When the scanning tip is moved away from the trench, the PL intensity decreases and its maximum blueshifts to ≈ 742 nm (see PL spectra recorded at position 2 and 3, 50 nm and 200 nm away from the trench, respectively), thus corresponding to ordinary PL emission from the bare exciton (see Supp. Fig. S1b). This behavior suggests that plexciton emission is strongly localized at the nanotrench as it is only detected when the probe is directly positioned over the trench, showing a peak wavelength that is consistent with the LP dip in the far-field reflection spectrum. We emphasize that LP plexciton emission is substantially stronger than emission associated with the surrounding uncoupled exciton. While the signal collected by the pyramidal probe comes from a region of approximately 300 nm in diameter[35], the plexcitonic emission signal originates from a much smaller area near the nanotrench, as confirmed below. This strongly enhanced, localized near-field emission allows us to spectrally resolve the near-field PL from the plexciton with sub-diffraction spatial resolution.

## Plasmon–exciton dispersion diagram
In order to analyze the plasmon–exciton coupling strength, we simulate the nanotrench plasmon resonance across the TMD exciton

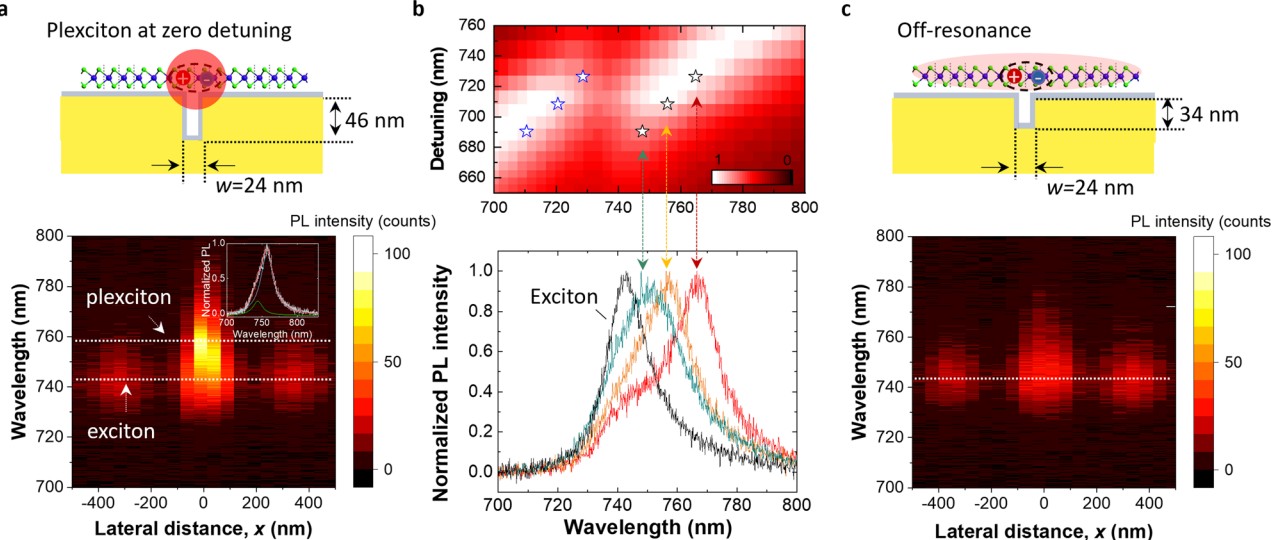

**Fig. 3 | Plexciton emission mapping and control experiment. a** Schematic view of a plexcitonic sample with a trench width of 24 nm and a depth of 46 nm, providing a zero detuning plasmon–exciton coupling. The bottom plot is the corresponding line scan analysis of PL spectra across the nanotrench, with the inset image showing the corresponding plexciton emission spectra deconvoluted with two peaks. **b** Upper panel: segment of the simulated dispersion diagram presented in Fig. 2d (color plot), alongside the observed LP positions in reflection (marked as stars) for three nanotrenches of different widths (different detuning wavelength). Lower panel: the corresponding normalized plexciton PL spectra revealing uncoupled excitonic emission and strongly coupled exciton with plasmon resonances at different detuning wavelengths shown in the upper panel; line scan analyzes of the green and red spectra with detuning wavelength slightly deviating from the zero detuning, along with reflection spectra, can be found in Fig. S27. **c**, Schematic view of the system for a gold nanotrench whose plasmon is off-resonance with the WSe$_2$ exciton (nanotrench with nominal $w = 24$ nm and $d = 34$ nm). The bottom plot is the corresponding line scan analysis of PL spectra across the nanotrench.

wavelength by tuning the trench width (see Supp. Sec. S3.C for details)[30,33,40,41]. The resonance of a single nanotrench is controlled by its width and depth and features a Fabry–Perot-like resonance analogous to the lowest energy surface plasmon polariton (SPP) mode of a metal–insulator–metal (MIM) waveguide traveling along the trench depth[41–43], corrected by the asymmetric truncation of the trench. In this picture, for a fixed nanotrench depth, narrower trenches are expected to exhibit plasmon resonances at larger wavelengths due to the stronger interaction between the SPPs at opposite trench walls. This intuition is supported by a comprehensive theoretical and experimental analysis of the dependence of the trench plasmon resonances on its geometry (width, depth, and shape) as well as the effect of the Al$_2$O$_3$ coating and PDMS residues (Supp. Sec. S4).

Before analyzing the coupled plasmon–exciton system, we first characterize the energy of the bare nanotrench plasmon and the uncoupled exciton of a WSe$_2$ monolayer on an unpatterned gold surface. The former is shown in Fig. 2a, where we identify a redshift of the plasmon resonance in the reflection spectra as the nanotrench width decreases, in good agreement with simulations (see Supp. Sec. S3). For example, the narrowest trench ($\approx 19$ nm wide) exhibits a clear reflection dip at $\lambda_p \approx 767$ nm; while for a width of 36 nm, the resonance wavelength moves to $\lambda_p \approx 642$ nm. This broad wavelength range allows us to tune the plasmonic resonance to sweep across the intrinsic exciton wavelength in the WSe$_2$ monolayer (742 nm, with a linewidth of $\approx 17$ nm; see Fig. 1d at 90° polarization and also Supp. Fig. S1b).

Next, we transfer WSe$_2$ monolayers to the fabricated nanotrenches and study the formation of LP and UP branches through reflection measurements. Again, our measurements reveal a clear avoided crossing pattern in the reflection spectra (Figs. 2b–c). The symbols in Fig. 2c indicate the spectral minima of the reflection curves for two sets of measured samples (filled circles and stars) featuring nanotrenches with slightly different depths (see also Supp. Fig. S13 for the influence of the nanotrench depth on the plasmon resonance). From these data, we estimate a Rabi-like splitting of about 80 meV,

which is comparable[20–23,44] or even surpasses[17,44] previous studies involving TMDs, including one in a similar configuration[17]. These observations are supported by simulations based on both the rigorous coupled-wave analysis (RCWA) technique[45–47] and the boundary-element method (BEM)[48] (see Supp. Sec. S5 for details). In particular, Fig. 2d shows the local density of optical states (LDOS) dispersion diagram calculated using BEM, to which we fit a coupled-oscillator model[1,7] to estimate the underlying plasmon–exciton interaction strength, yielding a Rabi-like splitting of 86.6 meV in excellent agreement with the experimental data (cf. Figure 2c). This value is also comparable to the strong-coupling criterion, defined by $\Omega > (\gamma_{plasmon} + \gamma_{exciton})/2$, given that our fitted exciton linewidth is $\hbar\gamma_{exciton} = 46.7$ meV (see Table S1 in Supp. Sec. S1.C) and the plasmon linewidth is $\hbar\gamma_{plasmon} \approx 130$ meV. Notably, removing the uncoupled exciton background results in an extracted Rabi-like splitting ($\Omega$) that is $\sim 10$ meV larger (see Supp. Sec. S5.A.1 and Fig. S24 for more details), further pushing our plexciton system just beyond the onset of the strong-coupling regime. In passing, we note that in our simulations we assumed 50-nm-deep nanotrenches instead of the experimentally estimated 46 nm $\pm$ 2 nm to account for a systematic redshift observed in the measured data. We attribute this shift to the presence of PDMS residues remaining from the TMD transfer process (see Supp. Sec. S4.E).

### Near-field probing of plexciton emission

Building on the far-field characterization of plasmon–exciton coupling presented above, we now employ a fiber-based pyramidal tip to investigate the near-field optical response of the plexciton by hyperspectral mapping of its PL emission at the nanoscale. Figure 3a shows the PL emission collected by the tip while scanning laterally across the nanotrench at zero detuning (i.e., when the plasmon resonance of the nanotrench matches that of the exciton in the TMD), revealing a pronounced near-field emission directly over the trench at a wavelength that matches the LP resonance observed in the far-field reflection spectrum. As in previous studies[14,22], we do not detect any observable

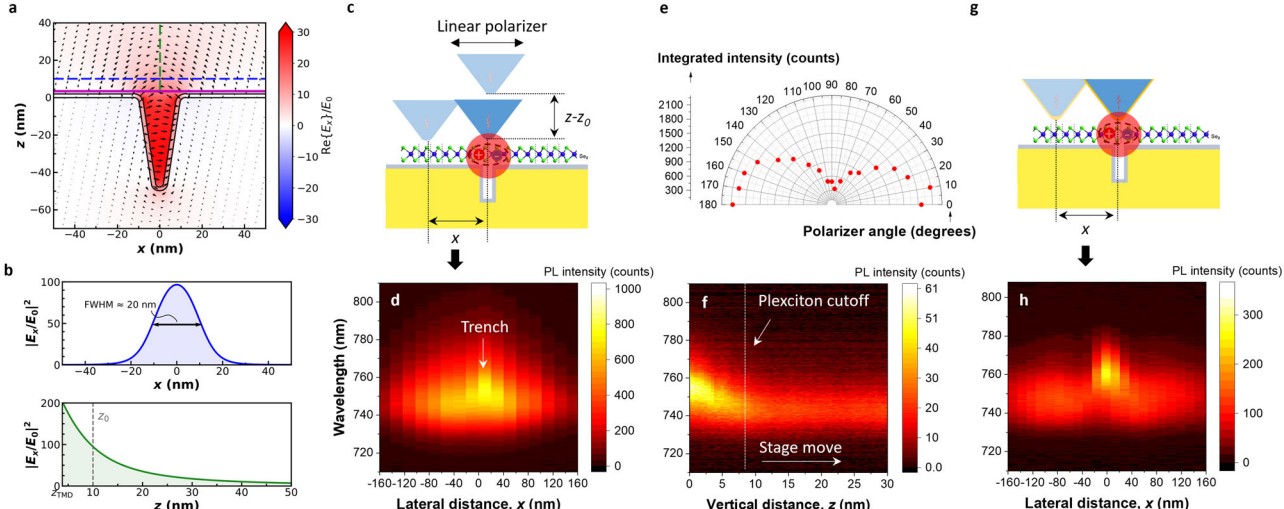

**Fig. 4 | Spatial localization and polarization properties of the plexciton.**
**a** Calculated electric field of the plexciton (LP) for a nanotrench corresponding to zero detuning ($w = 20$ nm and $d = 50$ nm). The vector field is represented by the arrows (length proportional to amplitude in log-scale). The background color plot shows the real-part of the $x$ component of the field. The ALD-coated nanotrench and the WSe$_2$ monolayer are outlined as black and purple lines, respectively.
**b** Corresponding field enhancement along the $x$ and $z$ directions [color-matching line cuts in (**a**)]. **c** Scheme depicting the scanning directions of the dielectric probe

relative to the nanotrench site along the lateral ($x$) and vertical ($z$) directions. **d** PL line scan along the lateral direction measured from a nanotrench at zero detuning ($w = 24$ nm and $d = 46$ nm) with a step size of 20 nm. **e** Polarization state of the plexciton assessed by rotating the linear polarizer at the output of the probe in increments of 10°. **f** PL spectra recorded as the probe is retracted from the $z_0$ position with a step size of 0.5 nm. **g** Similar to (**c**), but now employing a gold-coated probe. **h** PL map for a lateral line scan with a step size of 15 nm.

emission from the UP branch, presumably because it is quenched by nonradiative relaxation. To resolve the LP plexciton peak from the background, we fit the spectrum with two Lorentzians: one whose resonance and linewidth are fixed to those of the uncoupled emission peak (see Supp. Fig. S28) and another one with open fitting parameters. The result of this fitting procedure is presented in the inset of Fig. 3a, showing that plexciton emission is up to 5.75 times stronger than the background signal. This is notably stronger than reported in a previous study using scattering from a silver nanowire to readout the plexciton[49]. The fitted linewidth of the plexciton peak is slightly larger than that of the uncoupled exciton, which can be attributed to losses from the plasmon component in the plexciton.

Furthermore, we find that the measured near-field PL emission closely follows the reflection measurements of the lower plexciton branch as the plasmon resonance is detuned (Fig. 3b). This alignment is substantiated by comparing (i) a zoomed-in region of the calculated dispersion diagram (from Fig. 2d) with the LP dips in the measured reflection spectra superimposed, and (ii) the corresponding near-field PL emission spectra (see also Supp. Fig. S27). The latter are obtained by extracting and normalizing the plexciton spectra from the hyperspectral line scans (from Fig. 3a and Supp. Fig. S27a,b). Notably, the plexciton peak positions from these samples are located at the same LP absorption peak wavelength as indicated by the arrows, which confirms the correlation between the near-field plexciton peak and the LP detected in far-field reflection, which in turn confirms the formation of the plexciton at the trench position.

While detuning the plasmon resonance alters the plexciton emission, other factors could also locally modify PL from the TMD layer, such as local strain[50] and screening[51]. To exclude these possibilities, we designed and measured a control sample with an off-resonance plasmon mode. We maintained the trench width at 24 nm to ensure that the TMD layer experiences the same conditions over the trench while reducing the trench depth to 34 nm, thereby setting the plasmon resonance at 655 nm (see also Supp. Sec. S6.A). Reassuringly, the corresponding hyperspectral line scan only shows PL emission at the bare, uncoupled exciton wavelength with no spectral evidence of the plexciton. Unlike in the coupled system, there is no significant PL

enhancement over the nanotrench (Fig. 3c). Consequently, these observations rule out the effect of strain or probe-induced changes in the dielectric environment on the PL emission, further supporting our conclusions outlined above.

Having proven that the observed spectral signature is due to plexciton emission, we take advantage of this open-resonator platform combined with near-field characterization to directly probe, for the first time, the emission and field distribution of the plexciton. We leverage the capabilities of our near-field probe to investigate the spatial profile of the plexciton as well as the polarization state of its emission. Figure 4a illustrates the simulated electric field of the plexciton associated with the LP branch at zero detuning, showing that it is highly localized at the nanotrench and features a strong in-plane (across the trench) field component. Focusing on the area above the TMD (accessible by the near-field probe), we verify that the field is strongly confined both laterally and vertically. More quantitatively, we find a localization in $x$ with an FWHM of $\approx 20$ nm (same as the trench width) at $z = z_0$ ($z_0 \approx 10$ nm marks the initial scanning probe–sample distance[36]), and a vertical decay length within 20 nm of the TMD (see Fig. 4b and Supp. Fig. S25). We then experimentally examine the spatial profile of the plexciton, shown schematically in Fig. 4c, by scanning the dielectric probe along $x$ for fixed $z = z_0$ and also along $z$ by retracting the probe away from $z_0$ (Figs. 4d and f, respectively). Figure 4d shows a zoomed-in hyperspectral line scan (20 nm step size) across the trench close to zero detuning, featuring a strong emission enhancement centered around the trench and at a wavelength commensurate with the LP plexciton [linearly polarized across the trench (Fig. 4e), like the far-field data discussed above]. Incidentally, although the PL intensity is highest at the nanotrench's center, we still observe enhanced PL emission 100s of nm around the trench at the bare exciton wavelength, which is consistent with the optical resolution of the employed pyramidal probe[35]. We attribute this to weakly coupled (Purcell-type enhancement) excitonic emission. In contrast, the plexciton emission is exclusively observed when the probe is directly over the trench, with the PL peak shifting back to the bare exciton wavelength as the probe is moved as little as 20 nm away. Along the out-of-plane direction, PL emission from the plexciton could only be detected for probe

distances within $\approx 10$ nm from the initial probe−sample distance and is thus localized within $\approx 20$ nm above the TMD (Fig. 4f).

The observed plexciton localization, which is in agreement with the simulated spatial distribution of the plexciton electric field (cf. Fig. 4b and Fig. 4d, f), is confirmed using multiple near-field probes across different samples. This unprecedented confinement confirms that the far-field-detected strong coupling is highly localized at the trench position, where maximum field enhancement occurs. Additionally, the degree of spatial localization down to a few tens of nanometers is extraordinary compared to the diffraction-limited spatial resolution of the dielectric probe[35]. This exceptional resolution is attributed to the effect of scattering induced by the tip, with a spatial resolution comparable to its radius of curvature[52–54], or even a fraction of the tip radius[55,56]. Furthermore, this capability highlights the efficiency of near-field probes in detecting and analyzing localized plexciton emission. The ability to control and manipulate the polarization of excitonic emission through the formation of plexcitons adds a degree of freedom for tailoring the optical properties of 2D materials, enabling novel functionalities and potentially enhancing device performance.

Lastly, to better resolve the plexciton from the weakly coupled emission, we employed a pyramidal probe coated with a 20-nm-thick gold layer (Fig. 4g). Such a probe provides a more confined excitation and collection profile due to the larger field enhancement at the probe apex (with a 20 nm radius of curvature)[36]. This is demonstrated in Fig. 4h, which shows a hyperspectral line scan (15 nm step size) exhibiting stronger localization of the enhanced PL emission at the plexciton wavelength owing to the improved resolution of the metal-coated probe. However, we also detected a small redshift of about 5 nm in wavelength when probing the plexciton emission using this probe. This indicates that the gold coating slightly modifies the plasmonic mode of the trench, which justifies our choice in favor of the dielectric probe for a cleaner characterization. Nevertheless, we still observe a localized plexciton PL emission, and the improved resolution of a metal-coated tip could be advantageous in some settings. Looking ahead, the gold-coated tip could be leveraged to control plexciton emission−e.g., the associated light wavelength and/or polarization−as it can interact and modify the nanotrench plasmon resonance at small tip–sample separations. Additionally, the metal-coated tip opens the possibility of dynamically tuning the plexciton through the application of a bias voltage between the tip and the nanotrench[13].

In summary, we present a unique platform for investigating the nanoscale optical response of plexcitons by leveraging its capability for realizing strong plasmon–exciton coupling and consequent plexciton formation while also facilitating direct access to the plexciton near-field. In our plexcitonic device, the plasmon–exciton interaction is promoted by the large field enhancement with a strong in-plane component at the opening of a gold nanotrench provided by its plasmon resonance, thus ensuring large spatial overlap to match the dipole-moment of the 2D TMD exciton. Far-field reflection measurements indicate a substantial energy splitting of 80 meV at ambient conditions due to plasmon–exciton hybridization. In addition, we perform a comprehensive near-field study using a scanning probe that allows us to directly investigate the enhanced PL emission from the plexciton with a spatial resolution of a few tens of nanometers. We report enhanced near-field PL emission from the nanotrench site at wavelengths corresponding to the lower polariton resonance observed in the reflection spectra, supporting the conclusion that the detected emission originates from the lowest energy plexciton. This is further substantiated by measurements showing that the plexciton emission is strongly polarized across the trench direction. Our work opens new prospects for optical investigations of plexcitons and their emission dynamics at the nanoscale. Incidentally, while plexciton coupling in a silver nanoparticle on multilayer $WS_2$ has been mapped

with nanometric resolution using EELS[19], our SNOM-type approach not only constitutes a complementary avenue, but also offers several advantages and new capabilities. Besides benefiting from the high spectral resolution, flexibility, and integrability of optical setups, our technique facilitates polarization measurements and enables nanoscale mapping of plexcitonic PL emission, in contrast to EELS, which only provides information on absorption. Our technique has the potential to reveal key insights into relaxation pathways and can be paired with ultrafast lasers in a pump-probe scheme, thereby enabling future time-resolved studies. These attributes make the fiber-based SNOM technique introduced here particularly well-suited for current and future investigations of nanoscale polariton emission dynamics in complex nanostructures. We believe our work could fuel further research into the near-field optical response of plexcitons in hybrid metal–TMD architectures, with potential applications in nanophotonic devices, optoelectronics, and quantum electrodynamics in nanoscale cavities.

## Methods

### Sample preparation

**Gold nanotrenches and monolayer $WSe_2$.** The fabrication of the gold nanotrenches is described in Supp. Sec. S1.A. The exfoliation and transfer of the $WSe_2$ monolayers as well as their optical characterization is detailed in Supp. Secs. S1.B and S1.C.

**Nanoimprinted near-field fiber probe.** Diluted OrmoComp is dip-coated onto the end facet of a cleaved single-mode fiber (630HP) from Thorlabs. The fiber is mounted onto a piezo stage above the transparent mold and both the template and fiber end facet are visualized through the transparent mold. The mold fabrication is described in ref. 35, and it is reusable in our process. The fiber core is located by coupling red light into the fiber, then precisely aligned with the pyramid mold using a piezo stage. UV light (365 nm) is then illuminated through the template to expose the OrmoComp for a curing time of ~3 min.

### Optical measurements

**Reflection measurements.** The reflection spectra are measured using the Horiba CombiScope microscope equipped with custom-built optical components, including EQ-77 white light sources, an Andor Kymera 328i spectrometer, and a Thorlabs WP25L-UB broadband linear polarizer. For measurements on the $WSe_2$ monolayers and plexciton samples, we use a 100× objective lens with a 0.7 NA to focus the white light perpendicularly to the samples. For measurements of the resonance of the nanotrenches, we employ objectives with different magnifications and numerical apertures: a 10× objective with 0.25 NA for 1 μm pitch size, and a 100× objective with a 0.7 NA for the 680 nm pitch size. The reflection spectra are normalized as $R = R_{sample}/R_{substrate}$, where $R_{sample}$ is the reflectance measured from the sample (e.g., $WSe_2$ monolayer, bare nanotrenches, or $WSe_2$–Au-nanotrenches samples), and $R_{substrate}$ is the reflectance measured on the flat alumina-coated Au region adjacent to the sample.

**Hyperspectral PL mapping.** The scanning near-field optical microscope is based on the Horiba CombiScope microscope equipped with a shear-force head. The near-field probe is attached to a tuning fork oscillating at around 35 kHz, which modulates the tip-sample distance. The PL measurement are carried out in a fiber-in–fiber-out configuration. In this setup, a HeNe laser (633 nm) is coupled to the fiber using the Thorlabs MBT612D fiber launch stage, equipped with a 10× objective lens (0.25 NA). The emitted signal from beneath the probe is collected through the same path and directed to the spectrometer. As the tip scans across the sample, both the shear-force height information and the emission spectra are simultaneously recorded at each pixel.

## Theoretical modeling

**Dielectric function of WSe$_2$.** We fit the dielectric function of the WSe$_2$ monolayer using the reflection data obtained from an unpatterned region of the sample (Fig. S2). We describe the monolayer as an atomically thin slab with an effective thickness $t_{eff} = 6.49$Å (i.e., the interlayer spacing in the bulk material) and model its dielectric function in the spectral region of interest as a superposition of two Lorentzian terms as described in Supp. Sec. 1C. Among the fitting parameters (see Supp. Sec. 1C), we have obtained $\hbar\omega_1 \simeq 1.68$ eV and $\hbar\gamma_1 \simeq 46.7$ meV for the A-exciton frequency and linewidth, respectively.

**Optical response of gold nanotrenches.** The theoretical approaches for computing the optical response of the gold nanotrenches and the hybrid WSe$_2$–Au-nanotrenches structures are detailed in the Supporting Information. Hence, here we only provide a succinct description. In particular, we calculate the optical response of the involved nanostructures using two complementary approaches: one based on the RCWA[45,46] (using the open-source RETICOLO software[47]), where the sample is modeled as a one-dimensional grating; and another based on an in-house implementation of the BEM[48], where an individual trench is considered. In the BEM calculations, we consider V-shaped nanotrenches, as suggested by our experimental data (see Supp. Sec. 4.B for details) and reported in previous works[33,40,41].

## Data availability

The data that support the findings of this study are available within the manuscript and its supplementary information. Additional datasets and materials analyzed during the study can be provided by the corresponding authors upon request.

## Code availability

All codes used during the current study are either freely accessible or are documented in existing literature. The codes used in this study are available from the corresponding authors upon request.

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

## Acknowledgements
We thank our colleagues at the Molecular Foundry for their technical support and stimulating discussion. J. Z. thanks S. Sridhar and S. Virasawmy for their assistance on sample preparations. Work at the Molecular Foundry was supported by the Office of Science, Office of Basic Energy Sciences, of the U.S. Department of Energy, under Contract No. DE-AC02-05CH11231.

## Author contributions
J.Z., F.R., A.S., and A.W.B. conceived the project. J.Z. established the far-field and near-field optical capacities and performed the measurements. P.A.D.G. and F.J.G.A. developed the theory, and P.A.D.G. performed the calculations. S.D. performed the EBL. All authors discussed the results. J.Z. and P.A.D.G. wrote the manuscript with input from all authors. A.W.B. and F.J.G.A. supervised the project.

## Competing interests
The authors declare no competing interests.
