## [Transparent Peer Review file · Nature Communications]

Probing Plexciton Emission from 2D Materials on Gold Nanotrenches

Corresponding Author: Dr Junze Zhou

Version 1:

Reviewer comments:

Reviewer #1

(Remarks to the Author)

The manuscript "Probing plexciton emission from 2D materials on gold nanotrenches" presents a clever approach to induce strong coupling between the in-plane dipole moment of excitons in two-dimensional transition metal dichalcogenides (TMDs) and plasmonic modes through the nanotrench geometry. It provides one of the clearest demonstrations of strong coupling in 2D materials, including the observation of plexciton emission in photoluminescence, which is challenging to achieve in other nanoparticle-on-mirror systems due to quenching effects. Therefore, I recommend publication in Nature Communications after addressing minor comments.

1. The near-field probe without gold coating should effectively observe the intrinsic characteristics of plexcitons. Due to the strong in-plane field of the gold nanotrench, the polarization of plexcitons appears to be strictly confined to the in-plane orientation. While the authors mention the capability to control the polarization of plexciton emission, further clarification on how this controllability will be achieved is desirable. It is anticipated that the gold-coated probe might enable the manipulation of polarization, as the presence of nano-objects lifts wave vector conservation.

2. What is the origin of the confinement of plexciton emission? This phenomenon appears extraordinary, especially considering typical exciton diffusion and drift cases, and is notable even in the absence of bandgap engineering approaches.

3. The authors may consider citing a recent advancement in the control of plexciton emission [H. Lee et al. PRL 132, 133001 (2024)].

Reviewer #2

(Remarks to the Author)

In the manuscript titled "Probing Plexciton Emission from 2D Materials on Gold Nanotrenches", Zhou et al., studied both the far-field and the near-field optical response of strongly coupled plexcitons (plasmon-exciton polaritons) in a unique platform: WSe₂-covered gold nanotrenches hybrid device. The far-field reflection measurements are employed and demonstrate an energy splitting of 80 meV between plasmon and exciton. Moreover, taking advantage of the fiber-coupled SNOM technology, they directly map and access the nanoscale spatial distribution of plexcitonic excitations. The research topics related to the visualization of polariton in strong coupling regime is now pioneering and attractive in photo-physical science. However, the discussion here does not, in my opinion, convincingly confirm that measured near-field PL emission is from the LP branch of plexciton. It may well be, but further research is needed. I do not think that their expected impact on the studies of plexcitons warrants publication in Nature Communications which is targeted to a deeper physical understanding. Before next submission, I recommend the authors to consider the following points:

Firstly, the novelty of this paper is not as prominent as the author said. In fact, the nanometer-scale characteristics (spatial charge and field distribution) of TMDs-based plexciton has already been reported in Yankovich's work by using EELS in a scanning transmission electron microscope (Nano Lett. 2019, 19, 8171-8181). The used STEM EELS experiments have an unprecedented spatial resolution (<1 nm), which is more outstanding than the fiber-coupled SNOM technology in this work (a few tens of nanometers). If the direct spatial probe of plexciton is taken as the main point of the work, I think the authors should further state the unique advantage of their work (compared to Yankovich's work), rather than simply saying the SNOM technology is more cost-friendly, available and adaptable as SNOM is also not a common experimental setup in the Lab.

The width of the 1D gold nanotrenches used in the experiment is ranging from 19 nm to 36 nm (for the most used zero-detuning sample, the width is 24 nm). However, according to the manuscript, the spatial resolution of the SNOW technology is also around few tens of nanometers. Moreover, the spatial line scan step size described in Figure 4d to determine the plexciton field distribution is 20 nm. These featured length (both experimental spatial resolution and the scanning step size) is already compared to nanotrenches width itself. Therefore, I do not think the spatial distribution of plexciton in the x direction (20 nm based on the simulation) can be accurately characterized. To obtain a more convincing result, the authors are supposed to supplement the precise characterization of the spatial resolution of the SNOW system and the spatial characteristic size of the plexciton should be recalculated after considering the effect of Instrument Response Function. In the second part, the author recorded the reflection spectra of hybrid devices with different nanotrenches width and thus calculating the Rabi splitting of the system. (i) It is necessary for the authors to specify the spot size of white light used in reflection measurement and figure out whether uncoupled exciton signals (between two neighboring nanotrenches) will affect the extraction of strength of Rabi splitting. (ii) As referred in Nano Lett. 2023, 23, 444-450, The reflection spectra predominantly accessing the scattering signal of plasmon channel in coupled system, which might result in an overestimate of spectral Rabi splitting compared to absorption spectra, especially in case of intermediate coupling regime. Therefore, it would be more convincing for authors to utilize the absorption spectroscopy to characterize energy level splitting. (iii) The authors should verify whether plexcitons are valid formed in the device by using the criteria of strong coupling ($\Omega > (\gamma_{\text{plasmon}} + \gamma_{\text{exciton}})/2$). From my perspective, the device is indeed on the intermediate coupling regime (since the estimated linewidth is 130 meV for plasmon and 49 meV for exciton, so $(\gamma_{\text{plasmon}} - \gamma_{\text{exciton}})/2 < \Omega < (\gamma_{\text{plasmon}} + \gamma_{\text{exciton}})/2$). Therefore, the detected near-field PL signals (follows the reflection measurements of the lower plexciton branch) may not necessarily correspond to LP plexciton but could also be the Purcell-effect induced plasmon enhanced fluorescence.

In Figure 3b and Figure S25c, the authors found the measured near-field PL emission closely follows the reflection measurements of the lower plexciton branch as the plasmon resonance is detuned, and this finding is considered as the most direct evidence confirming the formation of plexciton in the hybrid device. However, according to their fitting results in Figure S26, the linewidth of LP peak is continuously reduced as the red tuning of plasmon resonance, which is obviously contrary to the luminescence property of plexciton. Because we know that with the red tuning of the plasmon resonance, the Hopfield coefficient of the plasmon component in LP will continue to increase, while the Hopfield coefficient of the exciton component will continue to decrease. This means the luminescence linewidth of the LP plexciton should increase with the redshift of plasmon resonance. However, the fitting-acquired linewidth decreases from 26.5 nm to 22.3 nm, and finally to 21.2 nm, which contradicts the theoretical results. Therefore, I do not think the near-field PL emission observed by the authors is from the plexciton emission. Probably, it is merely originated from the Purcell effect induced plasmon enhanced fluorescence.

Although the influence of local strain and discerning effect have been safely excluded by authors, the Shear force height map given in Figure S5 clearly shows the non-uniformity of WSe2 monolayer even on the same sample. Therefore, it is necessary for the author to further discuss the reproducibility of experimental results.

There is a minor markup error in the upper structure diagram of Figure 2b, the label Se2 should be replaced to WSe2.

Version 2:

Reviewer comments:

Reviewer #1

(Remarks to the Author)

The author has thoroughly answered all my questions and significantly enhanced the overall quality of the work. I have no further critiques.

Reviewer #2

(Remarks to the Author)

I would like to thank the authors for diligently addressing all my comments and suggestions provided for their manuscript. From my perspective, the authors' response, together with their revisions made to the manuscript, effectively improved the readability of the work and tackled most concerns raised. As a result, I am willing to say the manuscript is now ready for publication in Nature Communications and I hope the author can continue to consider two questions regarding Plexciton in their published version of this work.

1. I think the Hopfield coefficients (plasmon component) of LP peak of Plexciton should be necessarily principally monotonic in this context, which means the linewidth of Plexciton LP peak will continuously increase when plasmon resonance pass through the exciton resonance (from the blue-detuned state to the red-detuned state). Because plasmon component (large linewidth) will take more and more proportions of plexciton with less exciton component (small linewidth) remained. Thus, I am inclined to believe that the anomalous linewidth change is originated from the sample-to-sample differences. And I hope that, in the future work, the author can prove that the plexciton emission is effectively formed from the perspective of linewidth.

2. According to the cited reference [9] by authors, Plexciton is highly nonlinear and strongly interacting. The authors are supposed to simply supply a discussion on whether Plexciton nonlinearity (density) will affect their experimental results. More information related to plexciton nonlinearity can be referred in [K. Wei et al. Nat. Commun. 14, 5310 (2023)].

RESPONSE TO REVIEWERS' COMMENTS

Reviewer #1

Reviewer #1. The manuscript "Probing plexciton emission from 2D materials on gold nanotrenches" presents a clever approach to induce strong coupling between the in-plane dipole moment of excitons in two-dimensional transition metal dichalcogenides (TMDs) and plasmonic modes through the nanotrench geometry. It provides one of the clearest demonstrations of strong coupling in 2D materials, including the observation of plexciton emission in photoluminescence, which is challenging to achieve in other nanoparticle-on-mirror systems due to quenching effects. Therefore, I recommend publication in Nature Communications after addressing minor comments.

[Reply] We thank the reviewer for taking the time to evaluate our manuscript, for suggesting points to clarify in order to make the manuscript crisper, and for acknowledging both the scientific relevance and the novel approach we employed to overcome the longstanding challenge of measuring plexciton emission.

Reviewer #1. 1. The near-field probe without gold coating should effectively observe the intrinsic characteristics of plexcitons. Due to the strong in-plane field of the gold nanotrench, the polarization of plexcitons appears to be strictly confined to the in-plane orientation. While the authors mention the capability to control the polarization of plexciton emission, further clarification on how this controllability will be achieved is desirable. It is anticipated that the gold-coated probe might enable the manipulation of polarization, as the presence of nano-objects lifts wave vector conservation.

[Reply] The reviewer raises an important point. We present a novel plexciton system, where we characterize the intrinsic properties of the plexciton, namely, the spectral level splitting, the plexciton's spatial confinement, and the polarization of its emission. In particular, in terms of the plexciton's polarization, we are able to provide novel insights using our custom fiber-based dielectric probe. We demonstrate that the polarization state of the plexciton emission aligns with that of the nanotrench plasmon, which is linearly polarized across (i.e., perpendicular to) the nanotrench. This polarization state contrasts with that of the uncoupled excitonic emission, which is isotropic in the 2D material plane (and, thus, lacks a specific linear polarization). Our findings demonstrate that the formation of plexcitons is associated with well-defined emission characteristics that are substantially different from the bare, uncoupled excitons in the WSe₂ monolayer. This behavior is relevant as it enables the controlled readout of plexciton emission using a linear polarizer during collection.

In terms of the capability to control the polarization of plexciton emission, as originally mentioned in our manuscript, was indeed not clear and should be elaborated upon, as the reviewer correctly points out. In this work, the control over the (in-plane) polarization is limited to the degree of linear polarization, ranging from unpolarized (when the nanotrench plasmon is not resonant with the WSe₂ exciton) to linearly polarized across the trench (when the nanotrench plasmon resonance is tuned to that of the WSe₂ exciton). This is what we demonstrate. However, these results suggest that the polarization of plexciton emission could, in principle, be arbitrarily controlled using the polarization of the plasmon polariton (possibly with another type of plasmonic resonator). While the primary focus of this study does not center on manipulating the plexciton emission, this represents an interesting direction for future investigations. As noted by the reviewer, the gold-coated tip has the potential to control the plexciton—including its polarization—as it can interact and thus modify the nanotrench plasmon resonance*. Additionally, with the gold-coated probe, another enticing direction involves electrically tuning the exciton properties in 2D materials.

[Action taken] Following the reviewer’s comment, we have clarified this point in order to accurately reflect our findings by rephrasing the following sentence (at the end of paragraph 3, page 6):

“The ability to control and manipulate the polarization of excitonic emission through the formation of plexcitons emission adds a degree of freedom for tailoring the optical properties of 2D materials, (...)”

[Action taken] Stimulated by the reviewer’s remark on the gold-coated probe, and by the reference suggested in point 3 below, we comment on this possibility in our revised manuscript (added sentence in the last paragraph of page 6).

Looking ahead, the gold-coated tip could be leveraged to control plexciton emission—e.g., the associated light wavelength and/or polarization—as it can interact and modify the nanotrench plasmon resonance at small tip–sample separations. Additionally, the metal-coated tip opens the possibility of dynamically tuning the plexciton through the application of a bias voltage between the tip and the nanotrench.¹²

* In our case, the PL measurement using gold-coated fiber-tip (Fig. 4h) does not show any significant modification of the nanotrench plasmon resonance—besides a minor redshift of ~5 nm (this was further confirmed with numerical calculations)—because the tip was operating relatively far away from the nanotrench opening (>10 nm).

Reviewer #1. 2. What is the origin of the confinement of plexciton emission? This phenomenon appears extraordinary, especially considering typical exciton diffusion and drift cases, and is notable even in the absence of bandgap engineering approaches.

[Reply] The reviewer raises an excellent point. Indeed, we are visualizing a very strong localization of the plexciton. We clearly show reproducibly with our scanning near-field data on how the plexciton is highly localized at the nanotrench in both the lateral and the vertical directions. This level of localization, within a few tens of nanometers, is consistent with the curvature radius of our dielectric probe. Our hypothesis is that plexciton confinement is caused and dominated by the confinement of the plasmon polariton in the trench. The plasmon–exciton coupling that leads to the formation of a plexciton can only take place at the nanotrench where the plasmon polariton is localized.

Regarding the reviewer’s remark on exciton diffusion/drift, while excitons in TMDs can diffuse over 100s of nanometers during nanosecond timescales, however, long exciton diffusion/drift has been only reported for intrinsic (i.e., uncoupled) excitons in dielectric substrates or free-standing [PRB **90**, 155449 (2014), PRL **120**, 207401 (2018)]. Our case is substantially different because: the bare-exciton lifetime is expected to be reduced in our device due to the proximity of the metal substrate, thus limiting the diffusion distance; and, more importantly, plexcitons can only form in the region where the bare exciton and the nanotrench plasmon spatially overlap. Therefore, we do not expect exciton diffusion to be a significant contributing factor in our measurements.

[Action taken] We have added the explanation of the detect plexciton confinement and the spatial resolution in the 2nd paragraph of page 7.

The observed plexciton localization is in agreement with the simulated spatial distribution of the plexciton electric field (cf. Fig. 4b and Fig. 4d,f); is confirmed by multiple near-field probes across different samples. The unprecedentedly observed confinement confirms that the far-field-detected strong coupling is highly localized at the trench position, where the maximum field enhancement occurs. Additionally, the degree of spatial localization down to few tens of nanometers is extraordinary compared to the diffraction-limited spatial resolution of the dielectric probe.³⁴ This exceptional resolution is attributed to the effect of scattering induced by the tip, with a spatial resolution comparable to its curvature,^{51–53} or even a fraction of the tip radius^{54,55}. Furthermore, this capability highlights the efficiency of near-field probes in detecting and analyzing localized plexciton emission. ~~These results not only reveal that the plexciton is highly confined at the trench position but also that its PL emission can only be efficiently detected using a near-field probe.~~ (...).

Reviewer #1. 3. The authors may consider citing a recent advancement in the control of plexciton emission [H. Lee et al. PRL 132, 133001 (2024)].

[Reply] We thank the reviewer for bringing up this reference, which we regrettably missed and is indeed relevant in the context of our work.

[Action taken] We have now included it (reference 12 in the revised manuscript) in two different locations: one in the sentence (paragraph 2, page 1)

“(…) enable investigations of quantum phenomena at room temperature,^{1,2,11,12} (…)”

and also in the sentence added sentence that was previously mentioned in the point 1 above (i.e., related to the possibility of biasing the tip–sample when using a metal-coated tip).

Reviewer #2

Reviewer #2. In the manuscript titled “Probing Plexciton Emission from 2D Materials on Gold Nanotrenches”, Zhou etc., studied both the far-field and the near-field optical response of strongly coupled plexcitons (plasmon-exciton polaritons) in a unique platform: WSe₂-covered gold nanotrenches hybrid device. The far-field reflection measurements are employed and demonstrate an energy splitting of 80 meV between plasmon and exciton. Moreover, taking advantage of the fiber-coupled SNOM technology, they directly map and access the nanoscale spatial distribution of plexcitonic excitations. The research topics related to the visualization of polariton in strong coupling regime is now pioneering and attractive in photo-physical science. However, the discussion here does not, in my opinion, convincingly confirm that measured near-field PL emission is from the LP branch of plexciton. It may well be, but further research is needed. I do not think that their expected impact on the studies of plexcitons warrants publication in Nature Communications which is targeted to a deeper physical understanding. Before next submission, I recommend the authors to consider the following points:

[Reply] We sincerely thank the reviewer for their thorough evaluation of our manuscript and for raising several valuable points. These comments have allowed us to further substantiate our claims and enhance the overall clarity of our work. We are particularly grateful for the reviewer's recognition of our plexciton system as “a unique platform” and for acknowledging that our fiber-based tip can “directly map and access the nanoscale spatial distribution of plexciton excitations.”

Given the reviewer's expertise, we were initially surprised by the concerns regarding the evidence for near-field PL emission originating from the LP branch of the plexciton and the perceived lack of impact and novelty in our work. We realize now that our manuscript may not have sufficiently highlighted these aspects. In response, we have made significant revisions to the manuscript, including the addition of new measurements, as well as experimental and theoretical analysis. We believe these changes not only clarify the points raised by the reviewer but also substantially strengthen the manuscript.

Reviewer #2. Firstly, the novelty of this paper is not as prominent as the author said. In fact, the nanometer- scale characteristics (spatial charge and field distribution) of TMDs-based plexciton has already been reported in Yankovich's work by using EELS in a scanning transmission electron microscope (Nano Lett. 2019, 19, 8171-8181). The used STEM EELS experiments have an unprecedented spatial resolution (<1 nm), which is more outstanding than the fiber-coupled

SNOM technology in this work (a few tens of nanometers). If the direct spatial probe of plexciton is taken as the main point of the work, I think the authors should further state the unique advantage of their work (compared to Yankovich's work), rather than simply saying the SNOM technology is more cost-friendly, available and adaptable as SNOM is also not a common experimental setup in the Lab.

[Reply] Our work addresses two significant challenges in the field: (a) The creation of a strongly coupled plexciton by integrating a well-suited plasmon resonance with an exciton in a TMD *monolayer*. This leverages the substantially modified excitonic properties, such as oscillator strength, binding energy, and dipole orientation that emerge in the true 2D limit. (b) Granting direct access to the plexciton, enabling the mapping of its emission properties *at the nanoscale* for the first time, thus establishing a platform that could potentially harness the unique properties of plexcitons for device applications.

While the work by Yankovich et al. is indeed a significant contribution to the field (and for this reason it was already cited in our original manuscript), demonstrating the creation of a plexciton between an exciton in a layered semiconductor and a plasmonic nanoparticle, it is important to note that their study focused on a six-layer WS_2 , which is essentially equivalent to bulk WS_2 , rather than a monolayer. Moreover, although their characterization, via EELS, allows the mapping of polaritonic modes with nanometer resolution, it cannot address the light emission properties of the plexciton, an aspect that is central in our work. Therefore, while both our manuscript and the study by Yankovich et al. provide important insights, they inherently address different challenges in the field. It is thus unclear to us how the novelty of our work can be questioned by comparing it to the excellent, yet distinct, work by Yankovich and co-workers.

In addition, although the mentioned EELS study provided valuable information on plexciton coupling, it was not able to address some fundamental questions of plexciton formation in a true 2D system like the one considered in our work. Yankovich et al.'s EELS data could only show plexciton fingerprints in multilayer WS_2 (six layers, to be precise), which, in terms of band structure, resembles that of bulk WS_2 (featuring an indirect bandgap and thus indirect exciton formation), [A. Chernikov et al., Phys. Rev. Lett., 2014] not taking advantage of the sharp energy transition [L. Yuan et al., Nanoscale, 2015] and well-defined dipole orientation [J. A. Schuller et al., Nat. Nanotechnol., 2013] of the A exciton in a monolayer semiconductor. As a matter of fact, in their manuscript, the authors could not spectrally resolve the anticrossing behavior associated with the plexciton arising from the coupling between monolayer WS_2 and a Ag nanoparticle.

We clarify this point to avoid misunderstandings here. The work by Yankowich et al. is excellent and provided some critical legitimacy to study plexciton formation at room temperature using TMDs. However, here we were able to address insights that the community was seeking out,

specifically how to controllably couple a well-defined exciton in a 2D monolayer to a plasmon polariton, and how the emission pattern of such a 2D plexciton looks like. We addressed this challenge by engineering an entirely new platform that synergistically combines characteristics of excitons in a truly 2D semiconductor (strong exciton dipole moment with a well-defined polarization) with an appropriately designed plasmon resonance (overlapping with the TMD exciton spectrally, spatially, and in terms of polarization) to create a plexciton that can be directly accessed experimentally. Furthermore, we employ near-field optics to map, for the first time, emission and polarization properties of a 2D plexciton at the nanoscale, supported by theoretical calculations that capture the optical response of the system and plexciton properties in good agreement with the experimental data. Hence, we believe that we provide solid and timely insight into plexciton engineering with TMD-based systems.

[Action taken] To reflect these novelties, we have integrated the following extra text in the introduction (paragraph 2, page 1):

“Despite intense efforts to explore plasmon–exciton coupling in monolayer TMD/plasmonic hybrid systems, the nanoscale properties of plexcitons have so far been largely unexplored. The difficulty of accessing small plasmonic cavities and the limitations of conventional, diffraction-limited, photoluminescence (PL) setups have made probing these interactions extremely challenging.^{15,16} While electron energy-loss spectroscopy (EELS) has been employed to visualize strong coupling at the nanoscale, it was restricted to multilayer TMDs, where excitonic properties are less prominent compared to monolayers, and it could not access the emission properties effectively. Moreover, **S**trong plasmon–exciton coupling has been primarily realized by placing plasmonic nanoparticles onto TMD-covered substrates^{17–20} (including metallic ones, in the so-called nanoparticle-on-a-mirror setup²¹), making nanoscale characterization techniques based on scanning probes, such as scanning near-field optical microscopy or scanning tunneling microscopy, nearly impossible. **T**hus, **D**irectly accessing and probing a plexciton in the near-field has remained a pending challenge, yet is key to advancing our fundamental understanding of plexciton physics and to the purposeful design of plexcitonic devices.”

In addition, the reviewer’s comment motivated us to elaborate more on this point and to underscore the unique capabilities of our SNOM technique when compared with electron-beam-based ones. To that end, we have modified the second to last sentence in the Conclusion section as follows:

“Our work opens new prospects for optical investigations of plexcitons and their emission dynamics at the nanoscale, ~~creating a complementary technique to electron~~

~~microscopy,⁴⁸ with the added advantages of lower cost, wider availability, and greater adaptability of optical setups which are particularly suited to examine the emission properties of polaritons.~~ Incidentally, while plexciton coupling in a silver nanoparticle on multilayer WS₂ has been mapped with nanometric resolution using EELS,¹⁸ our SNOM-type approach not only constitutes a complementary avenue, but also offers several advantages and new capabilities. Besides benefiting from a high spectral resolution, flexibility, and integrability of optical setups, our technique facilitates polarization measurements and enables nanoscale mapping of plexcitonic PL emission, in contrast to EELS, which only provides information on absorption. Our technique has the potential to reveal key insights into relaxation pathways and can be paired with ultrafast lasers in a pump-probe scheme, thereby enabling future time-resolved studies. These attributes make the fiber-based SNOM technique introduced here particularly well-suited for current and future investigations of nanoscale polariton emission dynamics in complex nanostructures.”

Reviewer #2. The width of the 1D gold nanotrenches used in the experiment is ranging from 19 nm to 36 nm (for the most used zero-detuning sample, the width is 24 nm). However, according to the manuscript, the spatial resolution of the SNOW technology is also around few tens of nanometers. Moreover, the spatial line scan step size described in Figure 4d to determine the plexciton filed distribution is 20 nm. These featured length (both experimental spatial resolution and the scanning step size) is already compared to nanotrenches width itself. Therefore, I do not think the spatial distribution of plexciton in the x direction (20 nm based on the simulation) can be accurately characterized. To obtain a more convincing result, the authors are supposed to supplement the precise characterization of the spatial resolution of the SNOW system and the spatial characteristic size of the plexciton should be recalculated after considering the effect of Instrument Response Function.

[Reply] We thank the reviewer for the comment and suggestion. We recognize that the measured localization is directly at the spatial resolution of the SNOM and potentially appears even beyond its resolution. This level of localization also came to us as a surprise initially. As shown in the manuscript, we clearly demonstrate, in Fig. 3a and Fig. 4d, that the plexciton PL emission in the zero-detuning case is localized directly at the trench within the step-sizes of 50 nm and 20 nm, respectively. However, we reproduced this level of localization with a variety of near-field probes (reproduced with more than 10 different tips), including purely dielectric and Au-coated probes on multiple samples, and found this level of localized emission fully reproducible.

We believe that our high spatial resolution imaging can be attributed to the scattering from the near field into the far field, which has led us and other groups to report a spatial resolution consistent with or even below the typical radius of curvature of the probe. The scattering occurs

when a fraction of the tip radius protrudes into the plexciton near-field, scattering the signal into the far-field and resulting in an apparent resolution that is better than expected. This effect has been reported by the other groups, such as the Hillenbrand group for IR [S. Mastel et al., ACS Photonics, 2018] and ours for microwave near-field [M. I. B. Utama et al., Nat. Phys., 2020]. We draw this conclusion for three reasons: 1. The plexciton does not emit luminescence into the far field, and in the far field, we can only detect the plexciton in absorption. However, in the near field, we can detect the plexciton luminescence. This is a first indicator for scattering near field event, but not sufficient. 2. The luminescence can be detected using near-field probes without affecting the plexciton luminescence spectra. This is a very strong indication, especially using a dielectric probe, that we are effectively scattering from the plexciton near field into the far field. 3. We performed distance-dependent plexciton luminescence measurements with all different probes used and found that plexciton emission is undetectable when more than 10 nm away from the trench. This further supports the conclusion that the observed resolution is due to near-field to far-field scattering. Based on these observations, we believe that only a fraction of the tip radius protruding into the near field is sufficient to provide enough momentum to couple from the near field to the far field, allowing for spatial resolutions that are beyond the typical radius of curvature. Note, that in our case the resolution is just about or slightly below our radius of curvature, while in the case of super high resolution scattering SNOM the resolution can be even less than half of the radius of curvature.

Finally, we are also very conscious to not claim to have achieved 20 nm resolution, but rather indicate a “few tens of nanometers” of spatial resolution. In fact, achieving a spatial resolution lower than 20 nm with our SNOM probe is not anticipated, given that the curvature radius of our tip is around 20 nm. Nevertheless, we stress that our work is the first to map plexciton PL emission with such spatial resolution using an optical probe, by leveraging the capabilities of our fiber-based SNOM and by carefully designing a suitable plexciton system.

[Action taken] We have added our explanation of the achieved spatial resolution in the 2nd paragraph of page 7.

“The observed plexciton localization, which is in agreement with the simulated spatial distribution of the plexciton electric field (cf. Fig. 4b and Fig. 4d,f), is confirmed by multiple near-field probes across different samples. The unprecedentedly observed confinement confirms that the far-field-detected strong coupling is highly localized at the trench position, where the maximum field enhancement occurs. Additionally, the degree of spatial localization down to few tens of nanometers is extraordinary compared to the diffraction-limited spatial resolution of the dielectric probe.³⁴ This exceptional resolution is attributed to the effect of scattering induced by the tip, with a spatial resolution comparable to its curvature,^{51–53} or even a fraction of the tip radius^{54,55}. Furthermore, this capability highlights the efficiency of near-field probes in detecting and analyzing

localized plexciton emission. ~~These results not only reveal that the plexciton is highly confined at the trench position but also that its PL emission can only be efficiently detected using a near-field probe. (...).~~”

Reviewer #2. In the second part, the author recorded the reflection spectra of hybrid devices with different nanotrenches width and thus calculating the Rabi splitting of the system.

(i) It is necessary for the authors to specify the spot size of white light used in reflection measurement and figure out whether uncoupled exciton signals (between two neighboring nanotrenches) will affect the extraction of strength of Rabi splitting.

[Reply] We thank the reviewer for this comment. With respect to the beam size, we have verified that it is approximately 4 μm in diameter. From our analysis we convinced ourselves that neither the free excitons nor the pitch of the trenches affect the Rabi oscillation analysis. Firstly, note that the absorption signal due to uncoupled excitons amounts to less than 3% of the (relative) reflection, as illustrated in Fig. 1c (and reproduced in Fig. R1 below, for convenience) when the incident light is polarized parallel to the trench (refer also to the reflection measurement on the flat substrate region presented in Fig. S3a of the Supporting Information). This signal is significantly weaker than that from the coupled plexciton ($\sim 40\%$), and thus, the influence of the uncoupled exciton signal on the extraction of coupling strength is expected to be minor in our case. A second concern was the pitch between the trenches might interact with each other. Therefore, we modified the pitch and found no impact on the plasmonic resonance (see more description in suppl. S4A). Thus, while our far-field reflection signal comes from a few neighboring nanotrenches as well as from the area between nanotrenches, as the reviewer correctly pointed out, its effect on the splitted energy of the anti crossing is negligible.

Fig. R1 (reproduced from Fig. 1d in the manuscript): Reflection spectrum as a function of incident-light polarization (0° to 90° for perpendicular-to-parallel to the trench). Each curve is normalized relative to measurements on an unpatterned Al_2O_3 -Au substrate.

Nevertheless, prompted by the reviewer’s remark, we have performed additional calculations to more quantitatively assess the influence of the uncoupled exciton background on the extracted Rabi splitting. Figure R2 shows the reflection spectra for a zero-detuned sample and two orthogonal polarizations [across (TM) and along (TE) the nanotrench], along with the R_{TM}/R_{TE} ratio. Notably, under TE illumination, no trench plasmon is excited; thus, the spectrum is indistinguishable from that of a flat, unpatterned $WSe_2-Al_2O_3-Au$ structure ($R_{TE} = R_{flat}$). The nanotrench width, which defines the strongly coupled area, is much smaller (approximately 34 times) than the flat region. Therefore, by plotting R_{TM}/R_{TE} , we can effectively remove the contribution from the uncoupled exciton background (see Fig. R2). This analysis indicates that our previous results *underestimated* the Rabi splitting by ~ 10 meV due to this background, leading to a corresponding underestimation of the coupling strength amounting to ~ 5 meV (since $g \simeq \Omega/2$). Although this effect is relatively small, as discussed earlier, it may be non-negligible as it brings our system more into the strong-coupling regime [we revisit this point below, in the response to point (iii) from Reviewer #2].

Fig. R2: Reflection spectra from a zero-detuned sample (solid lines), showing the spectra corresponding to TM and TE polarizations, along with the ratio R_{TM}/R_{TE} , which effectively removes the uncoupled exciton background (see accompanying text). The arrows show the retrieved Rabi splitting base on both R_{TM} and R_{TM}/R_{TE} . The spectra were calculated using the RCWA method for a normally illuminated sample consisting of an array (of period $1 \mu\text{m}$) of 50-nm-deep nanotrenches.

[Action taken] We have added this new analysis to the main text [see comment (iii) below]. In addition, we have added Fig. R2 and an accompanying description to the SI (see Supp. Sec. S5.A.1).

Reviewer #2. (ii) As referred in Nano Lett. 2023, 23, 444-450, The reflection spectra predominantly accessing the scattering signal of plasmon channel in coupled system, which might result in an overestimate of spectral Rabi splitting compared to absorption spectra,

especially in case of intermediate coupling regime. Therefore, it would be more convincing for authors to utilize the absorption spectroscopy to characterize energy level splitting.

[Reply] We thank the reviewer for this suggestion, but we respectfully disagree with this statement. As a matter of fact, our reflection measurements (R) indeed give access to the absorption spectra (A), since the substrate is opaque and there is no transmission, and, thus, $R = 1 - A$. Our measurement configuration is conceptually similar to the absorption measurement approach described in the reference mentioned by the reviewer [Nano Lett. **23**, 444–450 (2023)].

However, it is important to note that the system studied in that reference involves few-layer MoSe₂ coupled to a plasmonic nanocavity within a nanocube on mirror (NCOM) geometry. This device differs significantly from our plexcitonic architecture in several key aspects. First, the NCOM platform exhibits distinct light scattering and absorption properties[†]. Additionally, the plasmon field in the NCOM configuration is confined in the gap region and oriented predominantly in the out-of-plane direction, which is less favorable for coupling with in-plane oriented TMD exciton. Lastly, the referenced study employs a *few-layer* TMD, whereas our system utilizes a *monolayer* TMD, further differentiating the two devices.

[†] For example, in the case of a metallic nanodisk on top of a TMD supported on a dielectric substrate, absorption dominates over scattering by more than an order of magnitude [see, e.g., ACS Photonics **6**, 994–1001 (2019), Ref. 20 in the original manuscript], which is exactly the opposite behavior to that observed in the NCOM-based study mentioned by the reviewer. We mentioned this here to underscore that the absorption/scattering efficiencies are highly system-dependent.

[Action taken] We have added the following description and cited the suggested reference in Paragraph 4, page 2.

“(…), we start by performing polarization-dependent reflection measurements by illuminating the sample with a white light of linear polarization varying from 0° (along the x -axis, perpendicular to the trench) to 90° (y -axis, parallel to the trench). The dips identified in the reflection spectra correspond to peaks in absorption,³³ which are typically observed in the far-field for strongly coupled systems. The results clearly show the presence of two well-resolved dips in the reflection data for 0° polarization, which progressively merge into a single dip at the bare exciton wavelength as the polarization is rotated to 90° (Fig. 1d). (...)”

Reviewer #2. (iii) The authors should verify whether plexcitons are valid formed in the device by using the criteria of strong coupling ($\Omega > (\gamma_{plasmon} + \gamma_{exciton})/2$). From my perspective, the device is indeed on the intermediate coupling regime (since the estimated linewidth is 130 meV

for plasmon and 49 meV for exciton, so $(\gamma_{plasmon} - \gamma_{exciton})/2 < \Omega < (\gamma_{plasmon} + \gamma_{exciton})/2$. Therefore, the detected near-field PL signals (follows the reflection measurements of the lower plexciton branch) may not necessarily correspond to LP plexciton but could also be the Purcell-effect induced plasmon enhanced fluorescence.

[Reply] We thank the reviewer for this comment. We should indeed explicitly gauge the observed Rabi splitting against the strong-coupling criterion $\Omega > (\gamma_{plasmon} + \gamma_{exciton})/2$. Given that $\hbar\gamma_{exciton} = 46.7$ meV (see Table S1 in Supp. Sec. S1.C) and $\hbar\gamma_{plasmon} = 130$ meV[§], we find that $(\hbar\gamma_{plasmon} + \hbar\gamma_{exciton})/2 = 88.35$ meV, which is comparable to the fitted Rabi splitting of $\hbar\Omega \approx 82$ meV and $\hbar\Omega \approx 87$ meV obtained through RCWA- and BEM-based calculations (see Figs. S21 and S22 in the Supp. Info., and also Fig. 2c,d in the manuscript), and consistent with mode splitting observed in our experimental reflection data. Hence, based on this analysis, our system is within the onset of the strong-coupling regime. Incidentally, removing the uncoupled exciton background [recall point (i) from Reviewer #2, and Fig. R2] leads to an extracted Rabi splitting ~ 10 meV larger, and, thus, pushing it further into the $\Omega > (\gamma_{plasmon} + \gamma_{exciton})/2$ range.

While our system is at the edge of strong coupling, we would like to reiterate that our near-field PL data clearly demonstrates that our system is in the strong-coupling regime. This is evident from the fact that the near-field PL emission peak recorded with our fiber-based SNOM occurs at the *exact wavelengths* associated with the lower polariton branch identified in our far-field reflection data. Such alignment cannot be explained by the Purcell-effect induced PL emission enhancement. If that was the case, the PL emission peak under zero-detuning conditions would align with that of the uncoupled exciton. Instead, our data reveal PL emission at longer wavelengths—see, for instance, the yellowish line in Fig. 3b, which corresponds well with the LP dip observed in the reflection spectrum. Moreover, a strong indication that our system is in the strong-coupling regime originates from the fact we could only detect PL emission from the low-energy plexciton, which is consistent with the formation of two new hybrid polaritonic levels and the quenching of the high-energy plexciton, as reported in previous studies (see Refs. 12 and 19 in the original manuscript).

Based on these considerations, we are confident that our near-field PL emission does come from strongly coupled plexciton emission.

[§] Our curve-fitting analysis yields bare nanotrench plasmon resonances with linewidths in the 120–140 meV range depending on the spectral window used for the fit, so we take the middle value of 130 meV as a representative one.

[Action taken] We have added the following strong coupling analysis to section 3 (paragraph 3, page 4):

“(…), yielding a Rabi splitting of 86.6 meV in excellent agreement with the experimental data (cf. Fig. 2c). This value is also comparable to the strong-coupling criterion, defined by $\Omega > (\gamma_{plasmon} + \gamma_{exciton})/2$, given that our fitted exciton linewidth is $\hbar\gamma_{exciton} = 46.7$ meV (see Table S1 in Supp. Sec. S1.C) and the plasmon linewidth is around $\hbar\gamma_{plasmon} \approx 130$ meV. Notably, removing the uncoupled exciton background results in an extracted Rabi splitting that is ~ 10 meV larger than the values stated above (see Supp. Sec. S5.A.1 and Fig. S28 for more details), further pushing our plexciton system just beyond the onset of the strong-coupling regime. In passing, (…).”

as well as section 4 (paragraph 1, page 6):

“Notably, the plexciton peak positions from these samples are located at the same LP absorption peak wavelength as indicated by the arrows, which confirms the correlation between the near-field plexciton peak and the LP detected in far-field reflection, which in turn confirms the formation of the plexciton at the trench position.”

Reviewer #2. In Figure 3b and Figure S25c, the authors found the measured near-field PL emission closely follows the reflection measurements of the lower plexciton branch as the plasmon resonance is detuned, and this finding is considered as the most direct evidence confirming the formation of plexciton in the hybrid device. However, according to their fitting results in Figure S26, the linewidth of LP peak is continuously reduced as the red tuning of plasmon resonance, which is obviously contrary to the luminescence property of plexciton. Because we know that with the red tuning of the plasmon resonance, the Hopfield coefficient of the plasmon component in LP will continue to increase, while the Hopfield coefficient of the exciton component will continue to decrease. This means the luminescence linewidth of the LP plexciton should increase with the redshift of plasmon resonance. However, the fitting-acquired linewidth decreases from 26.5 nm to 22.3 nm, and finally to 21.2 nm, which contradicts the theoretical results. Therefore, I do not think the near-field PL emission observed by the authors is from the plexciton emission. Probably, it is merely originated from the Purcell effect-induced plasmon enhanced fluorescence.

[Reply] We appreciate the reviewer’s comment regarding the linewidth of the plexciton emission. Concerning the remark on the exciton’s Hopfield coefficient, while the general argument is valid, it does not fully align with the specific cases presented in Figs. 3b and S25c. In these Figures, the spectra correspond to a blue-detuned sample, a zero-detuned sample, and a red-detuned sample, respectively. Consequently, the Hopfield coefficients of the plasmon and exciton are not necessarily monotonic in this context.

Based on our analysis, the observed decrease in linewidth from 26.5 to 21.2 nm is relatively minor and falls within the range of expected sample-to-sample variations. To illustrate this, we extracted and compared the PL spectra recorded between the trenches for the three samples (see Figs. R3a–c). The fitted Lorentzian profiles reveal slight variations in linewidth, with differences of a few nanometers. Similarly, the linewidth of the newly fitted plexciton peaks exhibits comparable variation (See Figs. R3d-f). Therefore, we attribute the observed changes in linewidth primarily to the sample-to-sample differences rather than to the detuning of the plasmon resonance[‡].

[‡] Incidentally, our numerical calculations predict slightly smaller linewidths of the bare nanotrench plasmon for trenches with resonances slightly to the red-side, which potentially could explain the behavior pointed out by the reviewer. Nonetheless, we reiterate that, since the observed linewidth variations are within the differences due to sample-to-sample variations (and even within different regions in the same sample), one cannot attribute a definite reason to the aforementioned *small* variations in the plexciton linewidth.

Fig. R3: Additional fits of the PL emission from different samples, containing both uncoupled exciton and plexciton contributions.

[Action taken] Reflecting this analysis, we have replaced Fig. S26 (now Fig. S28) by Fig. R3 and added a short description on the linewidth variation.

Reviewer #2. Although the influence of local strain and scattering effect have been safely excluded by authors, the Shear force height map given in Figure S5 clearly shows the non-uniformity of WSe₂ monolayer even on the same sample. Therefore, it is necessary for the author to further discuss the reproducibility of experimental results.

[Reply] We appreciate that the reviewer agrees with our conclusion that local strain and screening cannot explain our PL data. We also recognize the reviewer's concern about the potential effects of non-uniformities on plexciton formation. Hence, we were very careful in understanding and analyzing the potential non-uniformities:

1. Uniformity of nanotrenches: The uniformity of the nanotrenches directly impacts the plasmon resonance. We assessed this in two ways: (a) SEM data images of the patterned HSQ lines, which replicate the nano trenches, show uniformity within the resolution of SEM (See suppl. S4A). We did not directly image the replicated gold nanotrenched as the supporting material is photoresist which might be deformed during the e-beam exposure. (b) Optical response measurements were measured using a spot-size encompassing several trenches and were compared against numerical calculations with a fixed width. The consistency between the measurement and calculations, along with the comparable linewidths, indicates that inhomogeneous broadening is small and that the plasmon resonance is predictable.

2. Uniformity of the WSe₂ monolayer: The uniformity is evident from the consistent emission intensity recorded between the trenches, as depicted in Figs. S5a and S5b, as well as in the additional data presented in Fig. R4a, which is now included in the Supplementary Information.

3. The uniformity of WSe₂ over the trench. The shear-force height image presented in Fig. S5 was recorded with a step size of 50 nm, which is larger than the trench width. As a result, the probe may have “skipped over” the trench in some areas, failing to detect height changes at the trench positions. This results in an apparent nonuniformity, as seen in Fig. S5c. Additionally, PDMS residues on the sample surface could also further obscure the trench details, especially given the large step size. A similar “nonuniformity” is also observed in the new data shown in Fig. R4c, which also employs the same step size. This larger area scan shows the trench positions are not visible in some lines, corroborating the issues encountered in the previous measurements. To provide a clearer depiction of the trench topography, we have supplemented our data with images obtained using a cantilever-type sharp AFM probe (ATEC-NC) in Figs. R4b and R4d, which show the trenches with higher resolution.

Finally, we obtained data for the spectral splitting (Fig. 2), plexciton tunability (Fig. 3), and spatial localization of the LP branch (Fig. 4) in different samples and across two independent batches (indicated by the “stars” and “dots” in Fig. 2c of our manuscript).

With all the arguments and data presented above, we believe that our assumption of a fairly uniform sample is substantiated and the reproducibility of our results is robustly demonstrated.

Fig. R4: Additional data on the uniformity of the sample. (a, c) Correlated PL and shear-force height image. The white dashed line indicates the edge of the monolayer. (b, d) Phase and height images recorded using a cantilever-type AFM probe (ATEC-NC). The scalebars are 500 nm.

[Action Taken] Added Fig. R4 to the SI (after Fig. S5).

Reviewer #2. There is a minor markup error in the upper structure diagram of Figure 2b, the label Se2 should be replaced to WSe2.

Reply. We thank the reviewer for spotting that typo, which is now corrected (and the font enlarged for clarity) in our revised manuscript.

RESPONSE TO REVIEWERS' COMMENTS

Reviewer #1

Reviewer #1. The author has thoroughly answered all my questions and significantly enhanced the overall quality of the work. I have no further critiques.

Reply. We thank the reviewer for reviewing our manuscript as well as for all the provided comments which enabled us to further sharpen our manuscript.

Reviewer #2

Reviewer #2. I would like to thank the authors for diligently addressing all my comments and suggestions provided for their manuscript. From my perspective, the authors' response, together with their revisions made to the manuscript, effectively improved the readability of the work and tackled most concerns raised. As a result, I am willing to say the manuscript is now ready for publication in Nature Communications and I hope the author can continue to consider two questions regarding Plexciton in their published version of this work.

Reply. We thank the reviewer for the time taken to review our work, the constructive feedback, and for recommending our manuscript for publication.

1. I think the Hopfield coefficients (plasmon component) of LP peak of Plexciton should be necessarily principally monotonic in this context, which means the linewidth of Plexciton LP peak will continuously increase when plasmon resonance pass through the exciton resonance (from the blue-detuned state to the red-detuned state). Because plasmon component (large linewidth) will take more and more proportions of plexciton with less exciton component (small linewidth) remained. Thus, I am inclined to believe that the anomalous linewidth change is originated from the sample-to-sample differences. And I hope that, in the future work, the author can prove that the plexciton emission is effectively formed from the perspective of linewidth.

Reply. We once again thank the reviewer for this comment. We agree with the reviewer that, focusing on the lower polariton branch, the corresponding PL linewidth would be expected to continuously increase as the nanotrench plasmon resonance wavelength increases (i.e., crossing the bare exciton resonance from the blue-detuned to the red-detuned region), as the plexciton character evolves from exciton-like to more plasmon-like, with a concomitant increase in the plexciton's linewidth due to the increased plasmonic contribution (with an inherently broader

linewidth). This is indeed a valuable point that could be discussed in future studies. However, in our experimental data—as thoroughly detailed in the SI (see Sec. S6.C)—the observed variations in the linewidth of the plexciton emission peak are rather small. We attribute these variations to unavoidable sample-to-sample differences (as mentioned in Sec. S6.C), which is further supported by the fact that a similar variation of the linewidth is also observed in the bare, uncoupled exciton peaks from different samples. We stress, once again, that such minimal effects do not modify any of the conclusions presented in our manuscript.

Nonetheless, we appreciate the reviewer’s comment, which may inspire us to further explore this aspect in future studies.

2. According to the cited reference [9] by authors, Plexciton is highly nonlinear and strongly interacting. The authors are supposed to simply supply a discussion on whether Plexciton nonlinearity (density) will affect their experimental results. More information related to plexciton nonlinearity can be referred in [K. Wei et al. Nat. Commun. 14, 5310 (2023)].

Reply. We thank the reviewer for this comment. Our experimental setup utilized either a white light source (for reflection measurements) or a continuous-wave laser at relatively low power (for PL measurements), ensuring that all measurements were performed well within the linear regime. This is further supported by the excellent agreement with our linear-response theoretical calculations. Naturally, as the reviewer points out, our results would be affected under strong laser illumination using ultrashort laser pulses such as the ones employed in Ref. [9] and in K. Wei *et al.* Nat. Commun. **14**, 5310 (2023), due to, for example, modification of the dielectric function of the materials due to strong absorption, saturation effects, and the emergence of other exciton species (e.g., biexcitons or trions), just to name a few. Nevertheless, as mentioned above, these mechanisms are not pertinent to the current study, discussing them would only divert the readers' attention from the core messages and innovative aspects of our work.

Action taken. In response to the reviewer’s suggestion, we have included the suggested reference [K. Wei et al. Nat. Commun. 14, 5310 (2023)] in our paper (paragraph 2, page 1)

“(…) Plexcitons have been shown to produce large optical nonlinearities,^{9,10} (…)”